# Contributions to the Mathematical Modeling of the Threshing and Separation Process in An Axial Flow Combine

Nicolae-Valentin Vlăduț [1], Sorin-Ştefan Biriş [2,*], Petru Cârdei [1], Iuliana Găgeanu [1], Dan Cujbescu [1], Nicoleta Ungureanu [2], Lorena-Diana Popa [3,*], Lucian Perişoară [4], Gheorghe Matei [5] and Gabriel-Ciprian Teliban [6]

[1] National Institute of Research—Development for Machines and Installations Designed for Agriculture and Food Industry—INMA Bucharest, 013811 Bucharest, Romania
[2] Department of Biotechnical Systems, University Politehnica of Bucharest, 006042 Bucharest, Romania
[3] Agricultural Research and Development Station Secuieni, 617415 Secuieni, Neamt, Romania
[4] Faculty of Electronics, Telecommunications and Information Technology, University Politehnica of Bucharest, 060042 Bucharest, Romania
[5] Faculty of Agronomy, University of Craiova, 200585 Craiova, Romania
[6] Department of Horticulture, "Ion Ionescu de la Brad" Iasi University of Life Sciences, 700440 Iasi, Romania
* Correspondence: sorin.biris@upb.ro (S.-Ş.B.); dy.hemp420@gmail.com (L.-D.P.)

**Abstract:** The paper presents a mathematical model that characterizes the process of threshing and separation from the threshing machine with an axial flow of a thresher, taking into account the following input parameters: material flow, rotor speed, distance between rotor and counter rotor, mean density of processed material, feed speed, length of thresher and separating surface. Output parameters, such as the distribution function of separated seeds, distribution density function of separated seeds and distribution function of free seeds in the threshing space, as well as the distribution function of unthreshed seeds, together with the value of evacuation losses, were used to control the modeling process.

**Keywords:** modeling; threshing machine; seeds; separation

## 1. Introduction

With the advancement of bioengineering techniques, recent trends related to the crop harvesting process have focused mainly on mechanized crop harvesting, in order to significantly reduce the yield loss caused by manual harvesting, harvesting being a determinant of crop quality [1].

Grain harvesters can be classified as: combine harvesters and stationary threshing machines, with the mechanical methods of seed collection being based on impact, friction, and, respectively, simultaneous impact and friction [2–5].

The performances of the combine harvesters are influenced by different factors related to: crop characteristics (seed moisture, plant inclination and plant maturity), respectively, the characteristics of combine harvesters (forward speed, peripheral rotor speed, cutting machine speed, cutting height, shape and speed of the threshing machine, space between the rotor and the counter-rotor, type of threshing machine, sieve inclination and speed of air separation [6].

The main factor on which the evaluation of the performance of a combine harvester depends is the percentage of seed losses [7]. Seed losses resulting from harvesting operations can be divided into two main categories: pre-harvest losses (they occur before the actual harvesting process begins, being caused by insects, weeds, over-ripeness, birds and rodents), and losses due to the mechanical harvesting process (they occur between the beginning and the end of the combine harvesting process, being caused by seed damage, losses from cutting, threshing, separation and transport) [8]. Threshing losses involve the seeds damaged in the combine hopper and the seeds which were not separated from the

ears during threshing. Separation losses are seeds that have been separated from the ears, but are left behind the harvester [9,10]. In addition to the percentage of seed losses, combine harvesters are also assessed based on other indicators, such as the combined productivity, efficiency, energy consumption and decortication efficiency [1].

In addition to the natural and crop conditions, the quality and quantity of harvested seeds, the development of the working process of a threshing machine is also influenced by the physical-mechanical properties, including seeds' moisture, forward speed of the harvesting machine, material flow rate (entering the threshing machine), rotor speed, reel height (for the cutting stage), speed of rotation of the threshing machine, concave clearance, type of threshing machine, angle of inclination of sieves, frequency of sieves, air speed (for separation stage), conveyor screw rotational speed and conveyor clearance (for conveying stage) [11–17].

The main purpose of the mechanized threshing operation is to eliminate the manual execution of the process, reduce the threshing time, increase the quality of threshing seeds and increase threshing efficiency [18].

Experimental research has been conducted to investigate the operational factors (seed moisture, rotor speed and material flow) that affect the performance of a soybean unit with short axial flow, having a rotor with a diameter of 0.48 m and a length of 0.70 m, peg tooth clearance of 41.4 mm, concave clearance of 20 mm and guide vane inclination of 80°. The results demonstrated that the short axial-flow soybean threshing unit should be used for soybean seeds with moisture below 16% (as a lower percentage leads to a decrease in machine yield due to unthreshed seeds), rotor speeds are between 10–12 m·s$^{-1}$, while the value of the material flow is less than 150 kg·h$^{-1}$ [19]. It has been found that high moisture of rice straws and seeds affects the threshing process (rice variety being by nature difficult to thresh) [20,21]. When rice seeds have a moisture content of more than 24%, the percentage of unthreshed seeds increases as the rotor speed decreases, because the increase in rotor speed increases the resistance to threshing and therefore the loss of unthreshed seeds is reduced. Increasing rotor speed results in a reduced loss of unthreshed rice seeds when the moisture content is in the range of 20.6–32.1% [22,23].

Other studies have demonstrated that the percentage of broken seeds has increased when increasing the speed of the threshing machine, while increasing the moisture content of cereals and of unthreshed seeds, leading to its decrease [24,25]. Studies have also been performed to evaluate the influence of rotor speed on the efficiency of the threshing process [26,27]. Soybean threshing with axial flow thresher with higher rotor speeds (10.7–14.7 m·s$^{-1}$) increases the threshing efficiency [28].

Experimental research conducted on the threshing of sunflower seeds, influenced by feeding speed, rotor speed and type of threshing machine, demonstrated that seed losses in the threshing process or by impact can reach 6% [29]. Non-traditional methods of sunflower threshing depend on the phenomena of impact, friction and simultaneous impact and friction [30,31].

The performances of a belt threshing machine have been studied according to threshing efficiency, decortication efficiency, quantity of damaged seeds, machine productivity and specific energy consumption, taking into account three values of the pressure on the friction roller (2.0, 4.0 and 6.0 kg·cm$^{-2}$), four speeds of the friction roller (2.8, 3.7, 4.9 and 6.9 m·s$^{-1}$), different radial curves of the pressure surface (330, 345 and 365 mm) and different resting times of sunflower seeds inside the threshing chamber (5, 10 and 15 s) [32].

The paper presents a mathematical model that characterizes the process of threshing and separation from the threshing apparatus with an axial flow of a combine, taking into account the following input parameters: material flow, rotor speed, distance between rotor and counter rotor, mean density of processed material, feeding speed, length of the threshing apparatus and separating surface of the threshing apparatus with axial flow.

## 2. Materials and Methods

*Study Hypotheses*

The development of the working process of a threshing apparatus is strongly influenced by the physical-mechanical properties of the processed material, which depends on crop type and variety, pedo-climatic conditions of growth and development, and harvesting conditions. During the work, the cutting height of plants varies continuously, as the height of field is not always uniform, respectively; the combine follows the unevenness of the terrain, which is never flat after soil processing. This means that for the harvested material, the ratio *i* between seeds and straws and the values of straw content $\lambda$ or seed content $\nu$, used mainly in the calculations regarding the working capacity of the combine, vary within certain limits.

It is considered that:

$$i = \frac{S}{P} \tag{1}$$

$$\lambda = \frac{P}{S + P} \tag{2}$$

$$\nu = \frac{S}{S + P} \tag{3}$$

where: *S*—seeds; *P*—straw parts.

For a certain amount of material, the variation of these characteristics in relation to the actual threshing time is negligible [33], and therefore:

(a)  Ratio $S \cdot P^{-1}$ is considered constant;
(b)  The material is considered approximately homogeneous when feeding the threshing apparatus, respectively, the ears are evenly distributed in the mass of straw parts and material density is approximately the same over the entire width of the feeding surface.

Feeding uniformity is an important factor that appreciably influences the separation capacity of the threshing machine, and the hypothesis regarding the uniformity of material feeding means that feeding is uniform over time and also over the entire width of the threshing machine. It is considered that:

(c)  Plant material is introduced into the threshing machine and moves inside it as a continuous layer;
(d)  Slip resistance of the material in the threshing apparatus floor is a combination between the dynamic friction and mechanical interaction.

The mechanical interaction is dependent on the material, construction of the rotor and counter-rotor, as well as the relative speed between material layers; the threshing coefficient cannot be determined precisely in this case; thus, a constant mean value is adopted for it.

The seeds are detached from the ears primarily due to the impact with the rotor rails and then due to the combined actions of the compression, friction, tensile and bending forces of the ears, produced by the rotor and counter-rotor. Once detached from the ear, the seeds become free in the threshing apparatus floor. For the case studied in this paper, namely beater B 90 with axial flow (Figure 1), the following hypotheses can be made:

(e)  Seeds move in the space between the rotor and the housing, until they separate, with the same speed as the mixture of straw parts and unthreshed ears;
(f)  In the threshing apparatus floor, the material is homogeneous in a given radial section;
(g)  The mass of the material is continuously distributed in the threshing apparatus floor;
(h)  The density of a material volume element varies continuously from the entrance to the threshing apparatus to the exit, due to the separation of threshed seeds, compression and crushing of straw and variation of material speed (the flow of feeding material is considered constant).

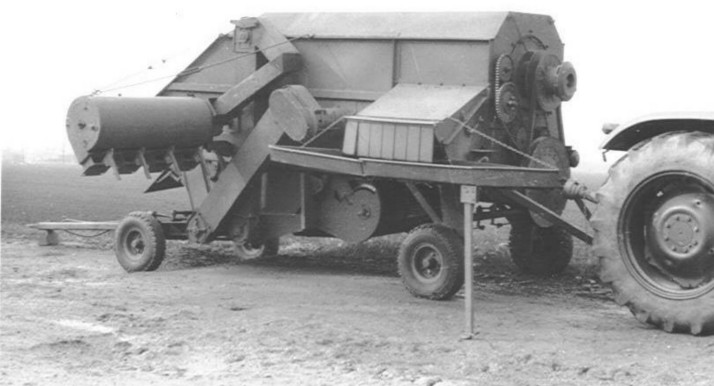

**Figure 1.** Perspective view of beater B-90.

The main adjustable and measurable parameters (influencing the performance of the work process) that were taken into account, were:

- Rotor speed $n$, adjustable within 620–1320 rpm; corresponding to this speed range, the peripheral speed of the rotor was in the range of 19.5–41.46 m·s$^{-1}$;
- Material flow $q$ [kg·s$^{-1}$], was determined by weighing the plant material sample and the time when the uniform feeding of the threshing apparatus was achieved. The mass of material introduced into the apparatus was verified for each test with the mass of the components collected following the threshing process. During tests, the material flow corresponding to the threshing apparatus width was modified in the limits 0.5–4.0 kg·s$^{-1}$, the combine having a maximum material flow of 5–6 kg·s$^{-1}$;
- The distance $\delta$ between the rotor's rails and the counter-rotor is adjustable, measured in the direction of forwarding of the material. It can be varied thus: $\delta_i$ = 12–29 mm at the entrance and $\delta_e$ = 3–7 mm at exit;
- The material feed rate was varied within the limits: 0.06–0.50 m·s$^{-1}$;
- The material feed angle can vary within limits: 0–15°.
- Peripheral speed of the rotor vp: 38–41 m·s$^{-1}$

The material was harvested and bound manually in the form of sheaves with a diameter of 170–260 mm, stored in the trailer of a tractor and covered with a tarpaulin, so that during the one month, approximately, that the experiments lasted (threshing and collecting in bags), it would not be compromised. The separation of seeds from the straw and chaff and weighing them for each of the 50 collected samples of the 17 experiments took about three months; during all this time, the moisture of the seeds and the chaff slowly increased from 9.88% to 14.8 % in the seeds and from 9.95% to 17.2% in the straw parts.

The main parameter, related to the material, modified during the experiments was the $S/PP$ ratio, which varied within the limits of 1/1.61–1/2.385, this being primarily due to the lower or higher height at which the plants were cut in relation to the ground.

The materials and tools used for handling and measuring the morphological characteristics of the material were: sickle, metric frame, measuring tape, bags, plastic bags and tarpaulins of different sizes, and a special installation was used to separate the seeds from the chaff and straw-stand for determining the floating speed.

The following equipment was used for weighing and determining the moisture content of the material in all experiments: tipping scale, electronic balance, analytical balance and laboratory oven.

During the experiments, the values of several parameters were varied in turn, namely: peripheral speed of the rotor (rotor speed), material flow rate, feeding speed and the distance between the rotor and the counter-rotor.

The separated material was collected in 50 cardboard boxes, numbered from 1 to 50, which were divided into two blocks: the boxes numbered from 1 to 25 were placed in a large metal box and those from 26 to 50 in another, in order to be easier to handle and to not destroy the cardboard boxes (due to the small space between the block of boxes and the

grid of the counter rotor, approximately 3–10 mm). The two boxes made of 1 mm metal sheet were carefully placed in the support guides on the chassis of the threshing apparatus module of the B 90. The block of collecting boxes numbered from 1 to 25 collected more material from the threshing area (being placed in front), while the block of boxes numbered from 26 to 50 mainly collected the material from the separation area.

In order to collect the evacuated straw parts, a tarp and the fly-panels for driving the material were properly placed, so that they does not spread over the seeds.

The plant material necessary for the test was weighed and then placed in the feeding space, with the ears forward, relative to the direction of movement (Figure 2).

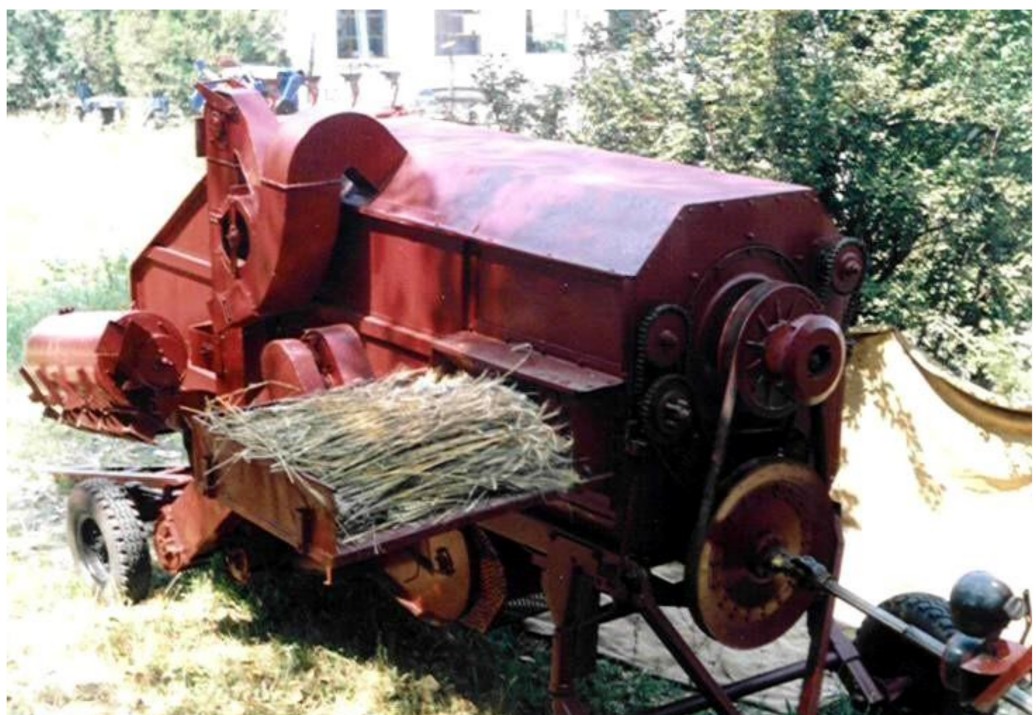

**Figure 2.** B 90 combine prepared for experiments.

The collection of the material pile separated by the counter-rotor was carried out with the help of a matrix (10 × 5) of collecting boxes, each box with the dimensions 200 × 200 × 100 [mm × mm × mm].

Figure 3 shows the construction diagram of the axial flow thresher. The counter-rotor, with a winding angle of 110°, has a construction similar to that of the usual tangential threshing machine, in the threshing area being placed three rails parallel to the rotor axis, and in the separation area; on the same generators on which the rails are mounted, the rotor is provided with three rows of separating plates, mounted inclined at an angle that can take the values of: 0°, 22.5° and 45°. The housing of the threshing apparatus is provided entirely with holes of 20 × 40 mm, the active separating surface of the housing representing about 55% of its total surface. Helical rails that are 30 mm high and 500 mm long are mounted on the sides of the housing, which can be mounted at different angles (60°; 75°).

Figure 3 shows a section through the threshing and separating apparatus of the axial flow beater B 90, under which the block of collecting boxes is mounted (Figure 4). Each row of boxes collects the material separately between two consecutive crossbars of the counter-rotor. The last line of boxes collects the material separated in the transition area towards the extension of counter-rotor. The collection of the discharged material at the exit of the threshing machine (fragmented straw, unthreshed ears, unthreshed seeds, chaff, etc.) was made on a 2 × 3 m$^2$ canvas. A tachometer was mounted on the rotor shaft in order to measure its speed at any time.

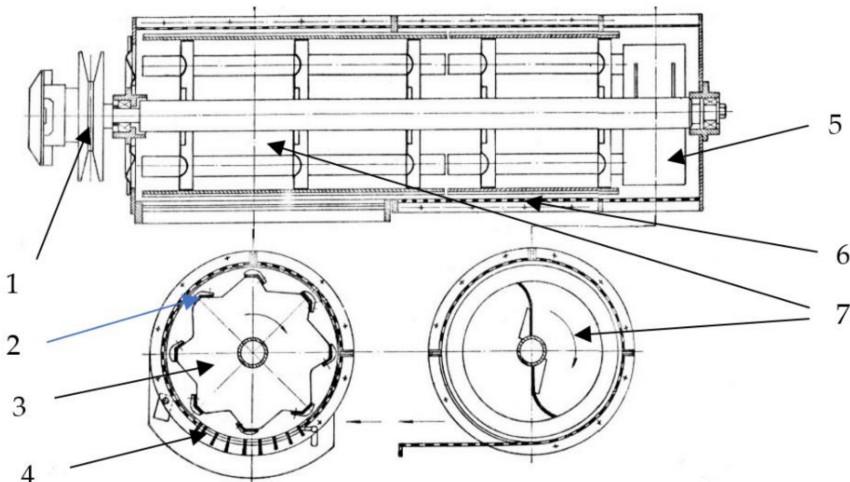

**Figure 3.** Section through the thresher apparatus of beater B 90, where: 1—threshing apparatus drive gin; 2—corrugated bar; 3—threshing apparatus rotor; 4—threshing apparatus concave; 5—material (ears) feeding section (area); 6—ears threshing section (area); 7—separation section (area).

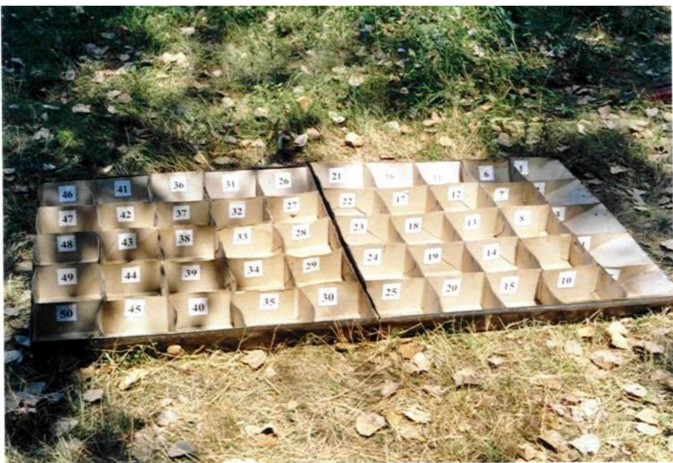

**Figure 4.** Block of collector boxes.

The axial threshing apparatus was fed tangentially, the feeding area of the rotor being equipped with rails. In the same area, the housing of the threshing machine is provided with spiral rails, mounted at an angle of 60° to the rotor axis, which has a diameter of 600 mm (length: 2 m). These rails are arranged on the housing at an angle of 180°.

The helical movement of the material in the threshing apparatus floor of the B 90 beater takes place due to the rails inclined on the rotor and the helical rails arranged on the housing.

The threshing machine performs the working process due to the action of the rifled rails and the counter-rotor bars on the vegetal material. Given the "$e$" hypothesis mentioned above, it can be considered that in the immediate vicinity of the rotor rail, a surface layer of material reaches a maximum speed, close to the peripheral speed of the rotor. The speed of the material layers decreases from the periphery of the rotor to the contact surface with the counter-rotor.

It results that those different forces act on the material found in the threshing apparatus floor, depending on the construction of the axial threshing apparatus, as well as on the instantaneous position of the material in the apparatus. Thus, in this type of axial threshing apparatus with tangential feeding, in the space between the rotor and the counter-rotor, material speed is generally tangentially oriented. In this part, the tangential speed of the material is similar to that of the tangential threshing apparatus.

In axial threshers with axial feed, in the space between the rotor and the counter-rotor, the helical movement is produced by the action of the inclined rails of the rotor, with both the rotor rails and the rails acting on the housing in the space between the rotor and the housing.

## 3. Results

### 3.1. Applying the Similarity Theory to Model the Threshing and Separation Process

For modeling the threshing process, the following physical quantities are set as input parameters of the process:

- $Q$—material flow [kg·s$^{-1}$];
- $n$—rotor speed [rpm];
- $\delta$—distance between rotor and counter-rotor [m];
- $\rho$—mean density of the processed material [kg·m$^{-3}$];
- $v_a$—feeding speed [m·s$^{-1}$];
- $L_t$—length of the threshing apparatus [m];
- $i$—seeds/straws ratio [$-$];
- $D$—rotor diameter [m].

It is considered, by a physically acceptable hypothesis, that there is a relation between some of these parameters:

$$Q = \rho \cdot S \cdot v_a \tag{4}$$

The material can be assimilated with a "fluid" consisting of several phases: seeds, straws, chaff, impurities, dust, air and possibly soil particles. The mean density of this "fluid" is denoted by $\rho$. In this context, if the size $S$ of the material intake surface is an adjustable or variable parameter from device to device, it should be included in the parameters list, but if it has a fixed value, then it can be removed from the discussion. For generalization, this parameter will be kept in mind next.

The "output" parameters of the process are: $S_d$—the distribution density of the separated seeds; $S_s$—separated seeds; $S_v$—damaged seeds; $S_l$—free seeds; $S_n$—unthreshed seeds; $p_{ev}$—evacuation losses; $\beta$—the separation coefficient (model parameter).

For the function of the distribution density of separated seeds, according to the hypothesis of author Miu [33], based on Lipkovich's model [34], the next equation is accepted for distribution density function, $S_d(x)$:

$$S_d(x) = \frac{\lambda \cdot \beta}{\lambda - \beta} \cdot \left( e^{-\beta x} - e^{-\lambda x} \right) \tag{5}$$

Moreover, for the function of the distribution of separated seeds, $S_s(x)$, the expression obtained by integrating the Relation (5) is accepted, whose primitive is of the form:

$$S_s(x) = \int_0^x S_d(u)du = \frac{1}{\lambda - \beta} \cdot \left( \beta \cdot e^{-\lambda x} - \lambda \cdot e^{-\beta x} \right) + 1 \tag{6}$$

However, for Equation (6) to be the distribution function of Expression (5) relative to the working interval [0, L], where L is the length of the distribution interval of the threshed material, the conditions that must be met are:

$$S_s(0) = 0 \tag{7}$$

And

$$S_s(L) = 1 \tag{8}$$

Condition (7) is fulfilled by the very Definition (6) of the function of distribution of separated seeds, while Condition (8) leads to the following connection between the parameters $\lambda$ and $\beta$:

$$\lambda = \beta + \frac{1}{L} \cdot ln\frac{\beta}{\lambda} \tag{9}$$

Therefore, the two parameters of the distribution density function of separated seed, $\lambda$ and $\beta$, are not independent, but are related to each other by Relation (9), which must be used when calculating the coefficients of unknown functions, using experimental data. It can be observed that, in order to make physical sense, the parameters have the size $L^{-1}$. The physical dimensions of the parameters involved in the process are given in Table 1.

**Table 1.** Physical dimensions of the parameters involved in the process.

| No. | Parameter | Exponent L (Length) | Exponent M (Mass) | Exponent T (Time) |
|-----|-----------|---------------------|-------------------|-------------------|
| 1 | $\lambda$ | −1 | 0 | 0 |
| 2 | $n$ | 0 | 0 | −1 |
| 3 | $\delta$ | 1 | 0 | 0 |
| 4 | $v_a$ | 1 | 0 | −1 |
| 5 | $\rho$ | −3 | 1 | 0 |
| 6 | $L$ | 1 | 0 | 0 |
| 7 | $Q$ | 0 | 1 | −1 |
| 8 | $S_S$ | 0 | 1 | 0 |
| 9 | $p_{ev}$ | 0 | 1 | 0 |

Thus: $\lambda = [L^{-1}]$; $n = [T^{-1}]$; $\delta = [L]$; $v_a = [LT^{-1}]$; $\rho = [ML^{-3}]$; $L_t = [L]$ and $Q = [MT^{-1}]$. There is a relation between the functions $S_s$, $S_l$ and $S_n$:

$$S_s(x) + S_l(x) + S_n(x) = 1 \tag{10}$$

The losses being given by the formula:

$$p_{tr} = \int_0^L S_n(x)dx, \, p_{sr} = \int_0^L S_l(x)dx, \, p_{ev} = \int_0^L (1 - S_s(x))dx \tag{11}$$

where: $p_{tr}$—threshing losses;
The relation between the distribution function $S_s$ and the distribution density $S_d$ is:

$$\frac{dS_s(x)}{dx} = S_d(x) \tag{12}$$

Which can also be written as [33]:

$$\frac{dS_s(x)}{dx} = \beta S_l(x) \tag{13}$$

where: $\beta$ is the separation coefficient $[L^{-1}]$.

As it can be observed from Table 1, the dimension of the considered dimensional space is $r = 3$, and the number of independent parameters considered is N = 7, i.e.,: $\lambda$, $n$, $\delta$, $Q$, $v_a$, $\rho$, $L_t$. Among the seven parameters listed above, there is a relation that connects three of them, namely the Relation (4). Therefore, one of these parameters can be removed from the following calculation. On the other hand, between the basic physical quantities, at least one of the parameters $Q$ or $\rho$ must be taken into account, otherwise the mass dimension is lost. Even if you work with the complete set of parameters and do not delete one with the help of Relation (4), this deletion can be performed even after the result of the dimensional analysis is obtained. According to theorem $\Pi$ of dimensional analysis, the phenomenon is characterized by a number of "N−r =4" dimensionless combinations and an F function, which connects them according to the relation:

$$\Pi_5 = F(\Pi_1, \Pi_2, \Pi_3, \Pi_4) \tag{14}$$

From the set of parameters that describe the phenomenon, we chose three fundamental parameters (whose dimensional determinant should be non-zero). Through tests, three fundamental parameters are chosen: $n$, $\delta$ and $Q$. For these parameters, the fundamental determinant is established (based on their coefficients):

$$\begin{bmatrix} 0 & 0 & -1 \\ 1 & 0 & 0 \\ 0 & 1 & -1 \end{bmatrix} = -1 \neq 0 \tag{15}$$

Given that the fundamental determinant is different from zero, it results that there is an interdependence relation between the three and then a dimensionless combination will be made next for each of the "N$-$3 = 4", remaining parameters: $\lambda$, $v_a$, $\rho$, $L_t$ and $S = Q \cdot v_a^{-1}$, depending on the three fundamental parameters previously chosen.

Therefore:

$$\begin{cases} \Pi_5 = \frac{\lambda}{n^{x_5}\delta^{y_5}Q^{z_5}} \\ \Pi_1 = \frac{v_a}{n^{x_1}\delta^{y_1}Q^{z_1}} \\ \Pi_2 = \frac{\rho}{n^{x_2}\delta^{y_2}Q^{z_2}} \\ \Pi_3 = \frac{L_t}{n^{x_3}\delta^{y_3}Q^{z_3}} \\ \Pi_4 = \frac{\frac{Q}{\rho v_a}}{n^{x_4}\delta^{y_4}Q^{z_4}} \end{cases} \tag{16}$$

where: $x$, $y$, $z$—exponents;

Each of the dimensionless, interdependent products is written in the dimensionless form of the function: $\lambda = [L^{-1}]$; $n = [T^{-1}]$; $\delta = [L]$; $v_a = [LT^{-1}]$; $\rho = [ML^{-3}]$; $L_t = [L]$ and $Q = [MT^{-1}]$. The corresponding dimensional equations will be:

$$\Pi_5 = \frac{\lambda}{n^{x_5}\delta^{y_5}Q^{z_5}} = \frac{L^{-1}}{T^{-x_5} \cdot L^{y_5} \cdot M^{z_5}T^{-z_5}} \Rightarrow L^{y_5} \cdot M^{z_5}T^{-x_5-z_5} = L^{-1} \cdot M^0 \cdot T^0 \Rightarrow$$
$$\Pi_5 = \frac{\lambda}{\delta^{-1}} = \lambda \cdot \delta$$

Since $\Pi_5 = F(\Pi_1, \Pi_2, \Pi_3, \Pi_4)$ and $\Pi_5 = \lambda \cdot \delta \Rightarrow \lambda = \frac{1}{\delta}F(\Pi_1, \Pi_2, \Pi_3, \Pi_4)$.

The same is applied for the product $\Pi_1 = \frac{v_a}{n^{x_1}\delta^{y_1}Q^{z_1}} \Rightarrow \Pi_1 = \frac{v_a}{n^{x_1}\delta^{y_1}Q^{z_1}} = \frac{v_a}{n\delta}$.

For product $\Pi_2 = \frac{\rho}{n^{x_2}\delta^{y_2}Q^{z_2}} \Rightarrow \Pi_2 = \frac{\rho}{n^{x_2}\delta^{y_2}Q^{z_2}} = \frac{\rho}{n^{-1}\delta^{-3}Q} = \frac{\rho n\delta^3}{Q}$.

Proceeding similarly, the other dimensionless products: $\Pi_3$ and $\Pi_4$ will result.

$$\begin{cases} \Pi_5 = \lambda \cdot \beta \\ \Pi_1 = \frac{\rho n\delta^3}{Q} \\ \Pi_2 = \frac{v_a}{n\delta} \\ \Pi_3 = \frac{L_t}{\delta} \\ \Pi_4 = \frac{\frac{Q}{\rho v_a}}{\delta^2} \end{cases} \tag{17}$$

From which the unknown exponents result $x_i$, $y_i$, $z_i$, where: i = 1, 2, 3, 4, 5. The final form of dependency is Equation (14):

$$\lambda = \frac{1}{\delta}F\left(\frac{\rho n\delta^3}{Q}, \frac{v_a}{n\delta}, \frac{L_t}{\delta}, \frac{\frac{Q}{\rho v_a}}{\delta^2}\right) \tag{18}$$

For simplification, the following notations are made:

$$p_1 = \frac{\rho n\delta^3}{Q}, \ p_2 = \frac{v_a}{n\delta}, p_3 = \frac{L_t}{\delta}, p_4 = \frac{\frac{Q}{\rho v_a}}{\delta^2} \tag{19}$$

From the concrete forms of function F, chosen during the analysis of the dependence of the experimental data ($\lambda$ as a function of $p_1, \ldots p_4$, in turn as a function of: $n$, $\delta$, $Q$, $v_a$, $\rho$, $L_t$) which appear in Table 2, following the use of the criterion of the minimum distance of the theoretical data from the experimental ones—$D_\lambda$ (dispersion) we chose the form from position 16. Other forms were examined but no more efficient functions than the one from position 16 were found. This analysis is conducted on an infinite number of possible functions. Thus, the concrete form of the dependence of function F on the parameters $p_1$, $p_2$, $p_3$ and $p_4$ (simplified form) is established by the authors, representing the dependence of the quantity $\lambda$ depending on parameters $p_1, \ldots p_4$ or their combinations. A similar dependency is obtained for parameter $\beta$.

**Table 2.** Variants of functions $\lambda$.

| No. | Parameter | Dispersion $D_\lambda$ |
|---|---|---|
| 1 | $38.502\frac{p_1}{\delta}$ | 0.20700013 |
| 2 | $0.547\frac{p_2}{\delta}$ | 0.22533057 |
| 3 | $0.004457\frac{p_3}{\delta}$ | 0.16572981 |
| 4 | $0.001041\frac{p_4}{\delta}$ | 0.17762885 |
| 5 | $\frac{29.05p_1+0.347p_2}{\delta}$ | 0.18200157 |
| 6 | $\frac{15.44p_1+0.003486p_3}{\delta}$ | 0.15471791 |
| 7 | $\frac{19.971p_1+0.0007714p_4}{\delta}$ | 0.15722568 |
| 8 | $\frac{-0.248p_2+0.005733p_3}{\delta}$ | 0.16040659 |
| 9 | $\frac{-0.091p_2+0.001152p_4}{\delta}$ | 0.17687420 |
| 10 | $\frac{0.012p_3-0.001789p_4}{\delta}$ | 0.15757224 |
| 11 | $\frac{-34.59p_1-0.479p_2-0.129p_3+0.016p_4+7.623}{\delta}$ | 0.14328398 |
| 12 | $\frac{13.446p_1-0.16p_2+0.004435p_3}{\delta}$ | 0.15254997 |
| 13 | $\frac{19.78p_1-0.053p_3+0.0008388p_4}{\delta}$ | 0.15693625 |
| 14 | $\frac{11.809p_1+0.006654p_3-0.0007215p_1}{\delta}$ | 0.15396032 |
| 15 | $\frac{-0.332p_2+0.015p_3-0.002225p_1}{\delta}$ | 0.14783308 |
| 16 | $3.135 \cdot 10^{-8}\frac{p_1 p_4{}^3}{\delta p_2{}^2} + 17.331$ | 0.12690374 |

*3.2. Considerations on The Density Function and the Probability Distribution of the Material Separated through the Space between Rotor and Counter-Rotor*

In Section 3.1, the suggestion to use for the function of material distribution density $S_d$, Relation (5) and implicitly for the distribution function $S_s$, Relation (6) was accepted. It has been observed that in order to satisfy the properties of a distribution function, for $S_s$, between the two parameters of Relation (5), Relation (9) must take place, which can also be written in the form:

$$L = \frac{ln\beta - ln\lambda}{\lambda - \beta} \tag{20}$$

This condition demonstrates that only one of the two parameters of Relation (5) is free. At this level, the work strategy bifurcates, with two ways of working being possible:

(1) Using the experimental data to determine the free parameter "A" so that the distribution function $S_s$ to model the experimental data "as well as possible", then to determine the dependence of this parameter on the process variables that appear in the argument list of the function F, for example in Equation (18); $A$ is a parameter that will be calculated by any process taking into account certain conditions of the process: $x = L \Rightarrow A = 1$.

(2) Considering the shape of the experimental curves, it is noted that the function $S_d$ generally has a global extremum in the working range [0, L], reference point for the experiment, and this model can be imposed on the modeling function, with some additional conditions.

Since the first path continues with the classical analysis of minimizing the function $F(\lambda, \beta, A)$ by the least squares method, being a classical path, the second path will be explained more.

$$F(\lambda, \beta, A) = \sum_{i=1}^{n} (S_s(x_i, A, \lambda, \beta) - S_s)^2 \tag{21}$$

If, taking into account the shape of the curves of experimental data, it would be necessary for the function $S_d$ given by Relation (5), to have the extreme coordinate point $(x_{S_{dmax}}, S_{d_{max}})$; then it is clear that the fulfillment of two more conditions besides Condition (9), by the two parameters $\lambda$ and $\beta$ of the function, is generally impossible, as the system is overdetermined.

In addition to these observations, it can be considered that Form (5) of the function $S_d$ cannot satisfy Condition (9) and the condition of having the above-mentioned extreme point at the same time, with the specification that $x_{S_{dmax}} \in [0, L]$. As the function must be continuous and derivable in the working interval, the abscissa of the extreme point must cancel the first derivative of the Function (5) in relation to the length of the thresher—$x$, hence the following relation is obtained:

$$x = \frac{ln\beta - ln\lambda}{\lambda - \beta}\bigg|_{S_{dmax}}, \tag{22}$$

This shows that the extreme point cannot be in the working range for Form (5) of the material density function.

Therefore, in order to exploit the second path of working (advantageous if the experiments are very precise, as the functions found in this way completely take over the experimental errors), the following alternative with three parameters is proposed, for the function $S_d$:

$$S_d(x) = A\left(e^{-\beta x} - e^{-\lambda x}\right) \tag{23}$$

By integrating the Function (23), it results in the expression for the distribution function:

$$S_s(x) = A\left(\frac{e^{-\lambda x} - 1}{\lambda} - \frac{e^{-\beta x} - 1}{\beta}\right) \tag{24}$$

The conditions for the function $S_d$ to have the extreme in the point $(x_{S_{dmax}}, S_{d_{max}})$, (indicated experimentally) and Condition (8), are materialized in the system of nonlinear equations (transcendent):

$$\begin{cases} \frac{ln\beta - ln\lambda}{\lambda - \beta} = x_{S_{dmax}} \\ -\beta x_{Sdmax}^{-\lambda x_{Sdmax}} \frac{e}{\frac{e^{-\lambda L} - 1}{\lambda} - \frac{e^{-\beta L} - 1}{\beta}} = S_{dmax} \end{cases} \tag{25}$$

If System (25) has a solution (the uniqueness problem remains to be solved), then parameter A is calculated according to the formula:

$$A = \frac{1}{\frac{e^{-\lambda L} - 1}{\lambda} - \frac{e^{-\beta L} - 1}{\beta}} \tag{26}$$

Notations:

$$p_1 = \frac{\rho n \delta^3}{Q}, \ p_2 = \frac{v_a}{n\delta}, \ p_3 = \frac{L_t}{\delta}, \ p_4 = \frac{S}{\delta^2} = \frac{\frac{Q}{\rho v_a}}{\delta^2} \tag{27}$$

With these notations, the functions established on the basis of experimental data can have any of the forms given in Table 2, determined by the least square's method using the Mathcad program. The functions of criteria 1, 2, 3 and 4 are partial linear functions of a single variable, and those of criteria 5, 6, 7, 8, 9 and 10 are partial linear functions of two variables. There is some superiority of the global approximation, 11. The most performant

rational formula is given at no. 16 in Table 2. Based on the experimental data obtained previously (during the tests), some approximate shapes of the allure of the curves that shape the work process can be identified. On these graphs, the functions from criteria 1÷16 were determined by successive tests. In order to select from the proposed combinations, the best one, the dispersion of the theoretical data compared to the experimental ones, was calculated.

Dispersion $D_\lambda$ (mean square deviation for $n$ values) is calculated on the graph of the obtained function as:

$$D_\lambda = \sqrt{\frac{\sum(x_i - \overline{x})^2}{n}} \tag{28}$$

According to the approximation criterion considered, the rational formulas from criterion 16 are the best. However, the search for accuracy may continue.

From Table 2, it can be noticed that the rational formulas from criterion no. 16 are the best ones, from the multitude of tested versions. After finding $\lambda$, $\beta$ is also determined in a similar way, which will have the form:

$$\beta = 44106.46 \frac{p_1{}^3 p_3}{p_2 p_4{}^2 \delta} + 2.76865 \tag{29}$$

And dispersion:

$$D_\beta = 0.02455466 \tag{30}$$

The formulas for $\lambda$ and $\beta$ are:

$$\begin{cases} \lambda = 3.135 \cdot 10^{-8} \frac{p_1 p_4{}^3}{\delta p_2{}^2} + 17.331 \\ \beta = 44106.46 \frac{p_1{}^3 p_3}{p_2 p_4{}^2 \delta} + 2.76865 \end{cases} \tag{31}$$

Using the notations (27), the formulas for $\lambda$ and $\beta$ can be written as follows:

$$\begin{cases} \lambda = 3.135 \cdot 10^{-8} \dfrac{\frac{\rho n \delta 3}{Q} \cdot \left(\frac{\frac{Q}{\rho \overline{v}_a}}{\delta 2}\right)^3}{\delta \cdot \left(\frac{v_a}{n \delta}\right)^2} + 17.331 \\[2em] \beta = 44106.46 \dfrac{\left(\frac{\rho n \delta 3}{Q}\right)^3 \cdot \frac{L_t}{\delta}}{\frac{v_a}{n \delta} \cdot \left(\frac{\frac{Q}{\rho \overline{v}_a}}{\delta 2}\right)^2 \cdot \delta} + 2.76865 \end{cases} \tag{32}$$

Having the values for $\lambda$ and $\beta$, the parameter $A$ can be calculated from Relation (26):

$$A = 3.37 \tag{33}$$

By introducing $A$ in Relation (24), we find the function of the distribution of the separated seeds, $S_s(x)$:

$$\begin{cases} S_s(x) = 3.37\left(\frac{e^{-\lambda x} - 1}{\lambda} - \frac{e^{-\beta x} - 1}{\beta}\right) \\ S_S(L) = 1 \end{cases} \tag{34}$$

From relations:

$$S_d(x) = A\left(e^{-\beta x} - e^{-\lambda x}\right) \tag{35}$$

$$\frac{dS_s(x)}{dx} = \beta S_l(x) \tag{36}$$

$S_l(x)$ is determined, thus:

$$S_l(x) = \frac{1}{\beta}\left(e^{-\beta x} - e^{-\lambda x}\right) \tag{37}$$

By introducing $S_s(x)$ and $S_l(x)$ in Relation (10), the function of the distribution of unthreshed seeds, $S_n(x)$, is also found:

$$S_n(x) = 1 - S_s(x) - S_l(x) \Leftrightarrow S_n(x) = 1 - A\left(\frac{e^{-\lambda x} - 1}{\lambda} - \frac{e^{-\beta x} - 1}{\beta}\right) - \frac{1}{\beta}\left(e^{-\beta x} - e^{-\lambda x}\right) \tag{38}$$

Having the values for $S_n(x)$, $S_l(x)$ and $S_s(x)$, from Relation (11), the evacuation losses $p_{ev}$ can easily be found:

$$p_{ev} = (1 - S_s(L)), \tag{39}$$

The aim of this modeling is to minimize the losses, in the conditions of achieving the best possible separation of the seeds from the ears, i.e., in a percentage of over 99%.

*3.3. Functions of Distribution ($S_s$) and Distribution Density ($S_d$) of Separated Seeds*

From the 16 variants of $\lambda$ functions (Table 2), for primarily physical and precision reasons (as shown by the precision estimators), the variant 16 was chosen. The form of this function, but also of the $\lambda$ obtained similarly, according to Relation (32), is:

$$\lambda = 3.135 \cdot 10^{-8}\frac{\rho n^3 S^3}{Q\delta^2 v_a^2} + 17.331, \beta = 44106.46\frac{\rho^3 n^4 \delta^{12} L_t}{Q^3 S^2 v_a} + 2.76865 \tag{40}$$

or (replacing $S = Q \cdot (\rho \cdot v_a)^{-1}$):

$$\lambda = 3.135 \cdot 10^{-8}\frac{n^3 Q^2}{\rho^2 v_a^5} + 17.331; \beta = 44106.46\frac{\rho^5 n^4 v_a \delta^{12} L_t}{Q^5} + 2.76865 \tag{41}$$

According to Relation (24), the function of the distribution of separated seeds has the form:

$$S_s(x) = A\left(\frac{e^{-\lambda L} - 1}{\lambda} - \frac{e^{-\beta L} - 1}{\beta}\right) \tag{42}$$

where:

$$A = \frac{1}{\frac{e^{-\lambda L} - 1}{\lambda} - \frac{e^{-\beta L} - 1}{\beta}} \tag{43}$$

The distribution density function is the derivative of the distribution function with respect to the variable $x$:

$$S_d(x) = A\left(e^{-\beta L} - e^{-\lambda L}\right) \tag{44}$$

Between the inlet flow, the density of the processed mixture, the feeding speed and the surface of the feeding nozzle, there is the Relation (4), which will be further taken into account: $Q = \rho \cdot S \cdot v_a$.

By using Relation (4), the flow can be eliminated from Relation (40). The expressions (45) of the parameters of the functions $\lambda$ and $\beta$ are thus obtained, which appear in the expressions of the distribution function and the distribution density function.

$$\lambda = 3.135 \cdot 10^{-8}\frac{n^3 S^2}{\delta^2 v_a^3} + 17.331, \beta = 44106.46\frac{n^4 \delta^{12} L}{S^5 v_a^4} + 2.76865 \tag{45}$$

It can be thus observed that the dependence on density $\rho$ disappears and that, if in general it can be considered $S$ = constant (due to the construction of threshing machines), then the parameters $\lambda$ and $\beta$ remain dependent only on n, $\delta$, $v_a$ and $L$.

*3.4. The Link between $S_s$ and $S_d$ Functions and the Experimental Results*

The determination of the coefficients of the parametric functions $\lambda$ and $\beta$, coefficients that appear in Relations (40) and (45) and in Table 2, was made starting from the experimental results that are given in the distribution function as a percentage. Therefore, in order to

be able to compare the experimental data with the theoretical results, some clarifications must be made.

To compare the distribution function directly in percentages, the experimental results are compared directly with the values of the function $S_s$ given in Relation (41), calculated for the given abscissa and multiplied by 100. If the comparison is to be made in terms of the mass of separate seeds (given in grams), then the total mass M of the separated seeds in the experiment is needed. The experimental values will be compared with the values of the $S_s$ function calculated in the given abscissa multiplied by the mass of separated seeds, M. These are the terms in which the experimental and theoretical results regarding the distribution function of the separate seeds are compared.

The distribution function $S_d$ represents, from a physical point of view, the quantity of seeds separated per unit length (meter). Basically, this function cannot be verified directly, as seed samples cannot be collected punctually. The experimentation procedure divides the length of the interval of interest, L, into $n$ intervals (for experiments in this material $n = 20$), harvesting and weighing the seeds separated from a box of length $L \cdot n^{-1}$ and of constant width. Therefore, if the division of the working range is:

$$\Delta_n = (x_0 = 0 < x_1 < \ldots < x_{n-1} < x_n = L) \tag{46}$$

Then the amount of separated seeds on the interval $(x_k, x_{k+1})$, is given by:

$$SD_{k+1} = \int_{x_k}^{x_{k+1}} S_d(x)dx \tag{47}$$

It is observed that:

$$SD_{k+1} = F(x_{k+1}) - F(x_k) \tag{48}$$

Therefore, the experimental data that appear as an expression of the distribution density function will be compared with the finite string $S_{d\,k}$. Moreover, these relations also result from the relation between the distribution function and the distribution density function in the experimental process, exactly determining the values of the terms of the finite string $S_{d\,k}$.

*3.5. Determination of the Other Functions' Characteristic of the Threshing and Separation Process*

In order to reach the final phase of the theoretical-empirical modeling of the threshing process, the phase of investigating the possibilities of optimizing the working process, the objective functions must be found. For this process, the main goal is to minimize losses. Losses at threshing and separation are the limit of the distribution functions of unthreshed seeds $S_n$ and free seeds in the threshing space $S_l$.

$$p_{tr} = \int_0^L S_n(x)dx, p_s = \int_0^L S_l(x)dx, p_{ev} = \int_0^L (1 - S_s(x))dx \tag{49}$$

In which the functions $S_n$ and $S_l$ are defined by Relation (10).

Relation (10) alone cannot provide both functions, $S_n$ and $S_l$, in paper [33], being given a relation that links the derivative of the $S_n$ function to the $S_l$ function. This relation is related to the form of the distribution function considered in the process (slightly different from the function considered in this paper); in the spirit of this relation, can be considered the relation:

$$\frac{dS_s(x)}{dx} = kS_l(x) or \frac{dS_s(x)}{dx} = \beta S_l(x) \tag{50}$$

However, since there is a relation between the distribution function of separated seeds and the distribution density function of separated seeds (12), there is a linear relation between the $S_l$ and $S_d$ functions:

$$S_l(x) = \frac{S_d(x)}{k} \tag{51}$$

Then, from Relation (10), it results that:

$$S_n(x) = 1 - S_s(x) - \frac{S_d(x)}{k} \tag{52}$$

The problem is solved if the constant $k$ is known. Thus:

$$\begin{cases} p_s = \frac{S_s(L)}{k}, \\ p_{ev} = L - A\left[\left(\frac{1}{\beta} - \frac{1}{\lambda}\right)L + \frac{e^{-\beta L}-1}{\beta^2} - \frac{e^{-\lambda L}-1}{\lambda^2}\right], \\ p_{tr} = p_{ev} - p_s \end{cases} \tag{53}$$

### 3.6. Solving of the Problem if a Type (50) Relation Is Accepted

The hypothesis in paper [33] is accepted partially, meaning that there is a linear dependency between the derivative of the distribution function of separated seeds and the distribution function of free seeds. In order to choose the proportionality constant $k$, a simplistic reasoning will be made, which will also highlight the limitation of the model. Both dimensionally ($[k] = L^{-1}$), and considering the hypothesis in [33], by analogy, we can consider $k = \beta$.

Accepting these hypotheses, the expression of function $S_n$ has the form presented in Relation (52). In order to verify the formulas obtained, the results will be calculated on a concrete case (experimentally obtained results), in which case the process parameters have the following values: $\rho = 77.78$ (kg·m$^{-3}$), $v_a = 0.09$ (m·s$^{-1}$), $n = 900$ (rpm), $\delta = 4.25$ (mm), $Q = 0.61$ (kg·s$^{-1}$), $S = 0.675$ (m$^2$) and $L = 2$ (m), respectively.

Based on the data from Relation (45), we calculated $\lambda$ and $\beta$, resulting in: $\lambda = 20.189$; $\beta = 2.769$. Then, introducing $\lambda$ and $\beta$ in Relations (23) and (24), it results that $A = 3.224$. Knowing $\lambda$, $\beta$ and $A$, we introduce in Relations (42) and (44), obtaining the equations for $S_s(x)$ and $S_d(x)$, as follows:

$$S_s(x) = 1.00462 - 1.1643e^{-2.769x} \tag{54}$$

$$S_d(x) = 3.224\left(e^{-2.769x} - e^{-20.189x}\right) \tag{55}$$

Under these conditions, the variation of the distribution function with abscissa $x$ has the form given in Figure 5, the values of the distribution function are those given by Formula (42), the percentage form are obtained by multiplying by 100, and the values in grams are obtained by multiplication by the total mass of the sample, with the deviation being 1.29%.

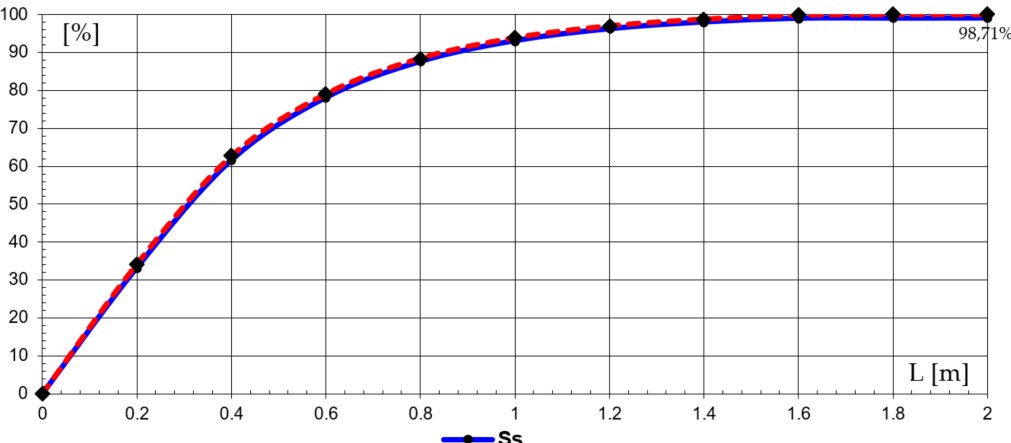

**Figure 5.** Variation of the distribution function of separated seeds $S_s$ along the length of the threshing apparatus L (where: ◆ experimental points; ▾▾▾▾▾▾ curve drawn by points; ▬▬▬ theoretical curve).

Under the same conditions, the distribution density function varies, as in Figure 6.

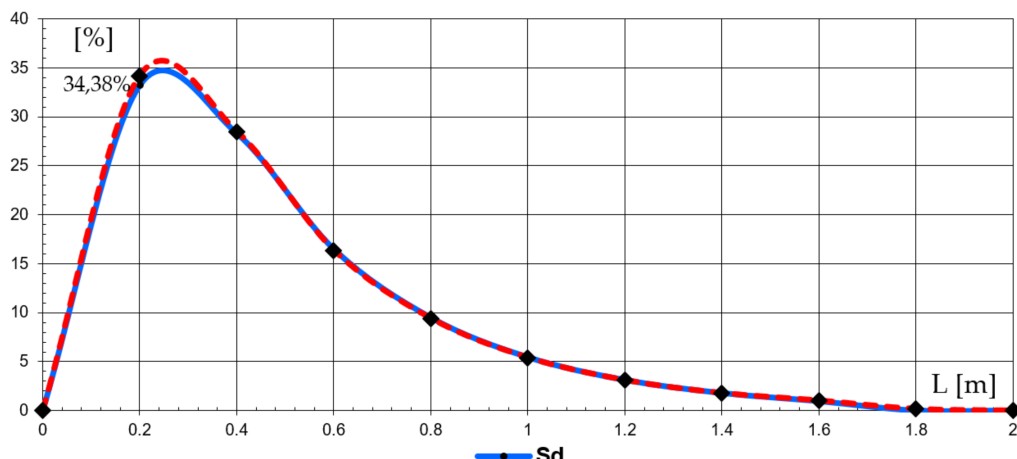

**Figure 6.** Variation of the distribution density function $S_d$ along the length of the threshing apparatus L (where: ◆ experimental points; ▪▪▪▪▪▪ curve drawn by points; ▬▬▬ theoretical curve).

The distribution function of the free seeds in the threshing apparatus floor varies with the length of the threshing machine L, as in Figure 7.

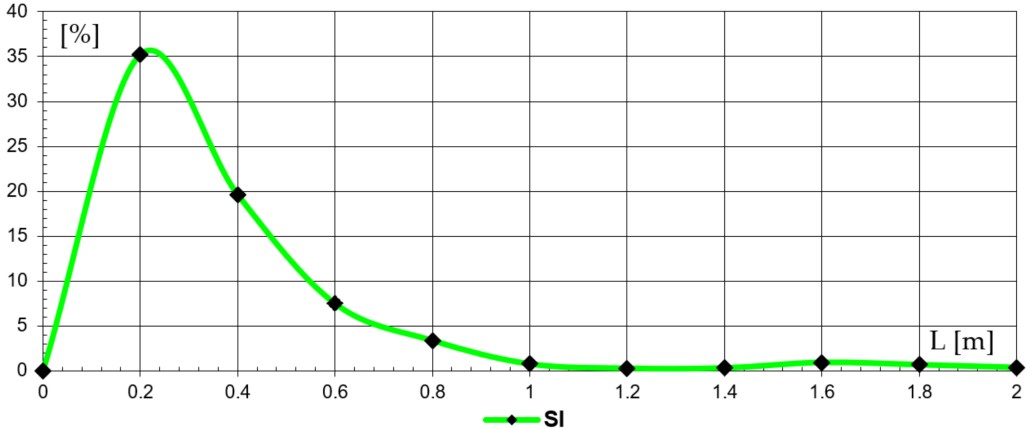

**Figure 7.** Variation of the distribution function of free seeds $S_l$ along the length of the threshing apparatus L (where: ◆ experimental points; ▬▬▬ curve drawn by points).

The variation of the distribution function of the unthreshed seeds with length L of the threshing apparatus is shown in Figure 8.

The variation of the distribution function of the separated seeds $S_S$, respectively, and separated seeds Sd with length L of the threshing apparatus, is shown in Figure 9, respectively, in Figure 10.

Using the data obtained with the help of the mathematical model, Figure 11 shows the variation of the percentage of separated, unthreshed, free seeds and of the distribution density depending on the rotor length of the combine, for experiments A14–A17 (where: $n = 900$ rpm = ct; $Q = 0.838$ kg·s$^{-1}$ = ct; $v_a = 0.13235$ m·s$^{-1}$ = ct).

It can be observed that the amount of free seeds in the straws has a high value in the first part of the threshing apparatus (at the beginning of separation), due to the fact that in the first area, the threshing operation is mainly performed, followed by more and more of a pronounced decrease in this amount when entering in the separation zone of the apparatus.

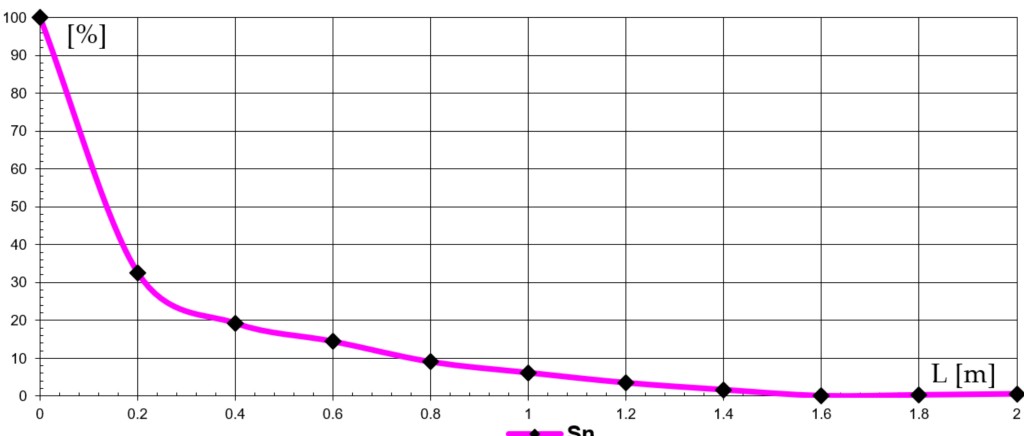

**Figure 8.** Variation of the distribution function of unthreshed seeds $S_n$ along the length of the threshing apparatus L (where: ♦ experimental points; ━━━ curve drawn by points).

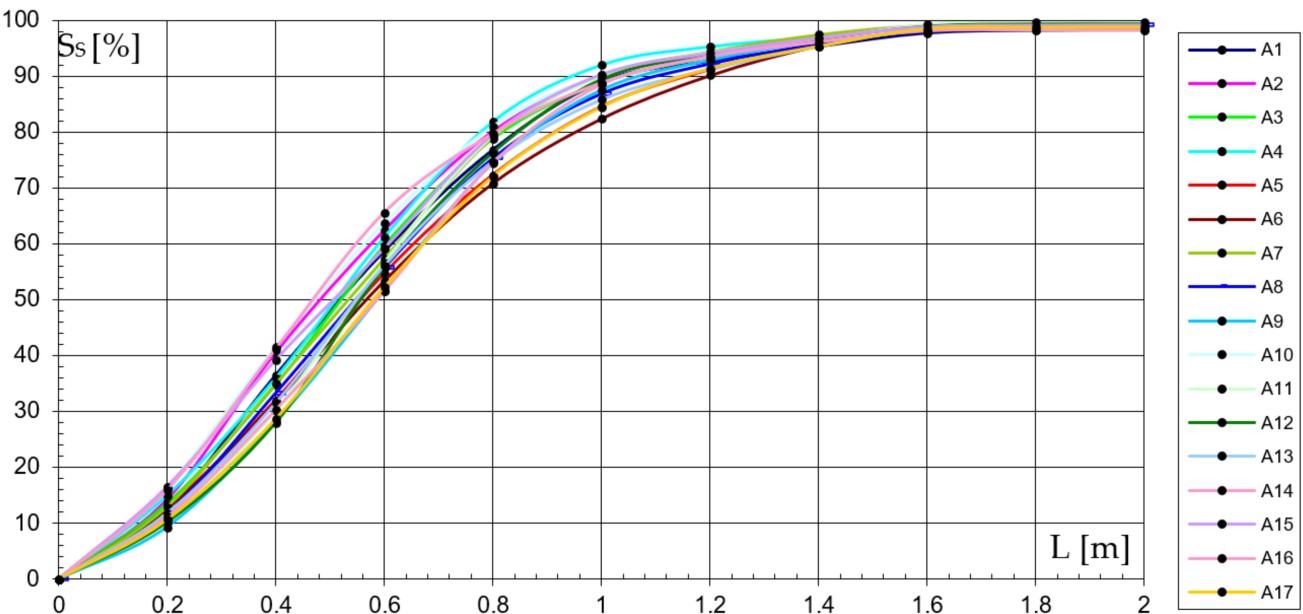

**Figure 9.** Variation with the length of the threshing apparatus L, of the experimental values (experiments A1–A17), for the function of distribution of the separated seeds $S_S$.

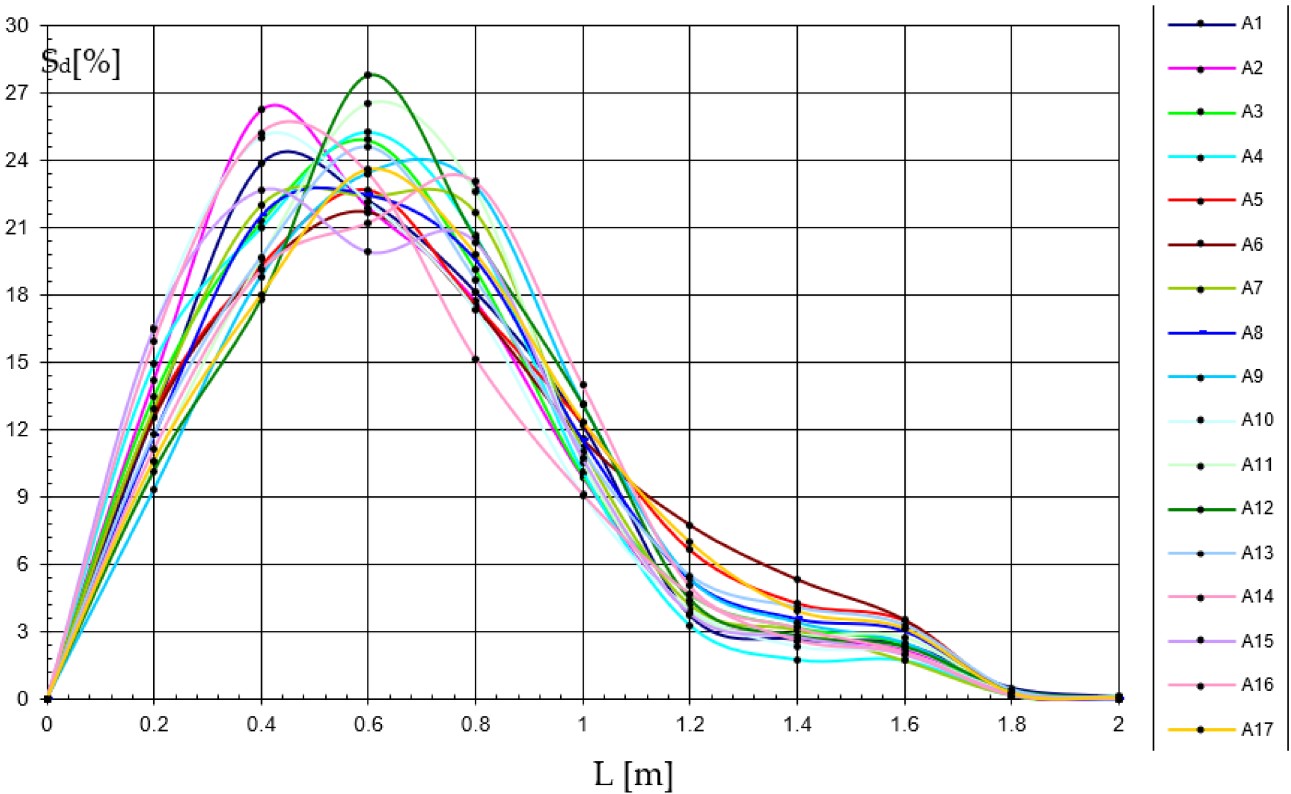

**Figure 10.** Variation with the length of the threshing apparatus L, of the experimental values (experiments A1–A17), for the density distribution function of separated seeds $S_d$.

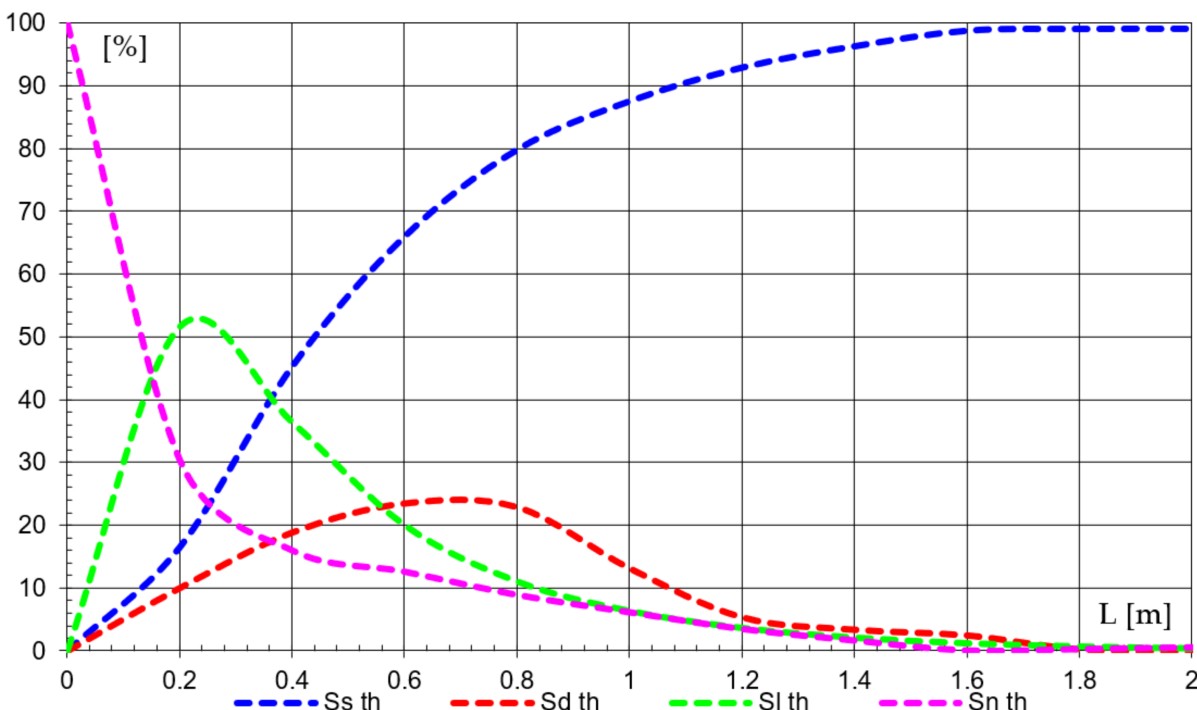

**Figure 11.** Variation of the percentage of separated seeds ($S_s$), unthreshed ($S_n$), free ($S_l$) and distribution density ($S_d$) along rotor length L (for the average of values from experiments A14–A17, determined theoretically) where: $n$ = 900 rpm = ct; $Q$ = 0.838 kg·s$^{-1}$ = ct; $v_a$ = 0.13235 m·s$^{-1}$ = ct).

## 4. Conclusions

Given that it is very difficult to develop a mathematical model for the threshing and separation process that takes into account all input parameters that directly or indirectly influence this process, in order to model the process well enough so that no major deviations occur from the real process, but not to complicate the model too much, for the realization of this modeling, only seven of the most important input parameters were taken into account: $Q$ (kg·s$^{-1}$), $n$ (rpm), $\delta$ (m), $\rho$ (kg·m$^{-3}$), $v_a$ (m·s$^{-1}$), $L_t$ (m) and $S$ (m$^2$).

In order to control the modeling process, what must be obtained (quantified) was established, i.e., the output parameters, in a first phase having: $S_s(x)$—the distribution function of separated seeds, $S_d(x)$—the function of density distribution of separated seeds, $S_l(x)$—the function of distribution of free seeds in the threshing space and $S_n(x)$—the function of distribution of unthreshed seeds; the determination of these parameters is followed by establishing the other output parameters that quantify the quality of the threshing and separation process: $p_{ev}$—value of evacuation losses.

In order to obtain a modeling as close as possible to the reality of the threshing and separation process, a dimensional analysis of the considered input parameters was performed, so that the dependence between the parameters was obtained, but especially the dependence relation between coefficients $\lambda$ and, respectively, $\beta$, depending on the considered parameters, and the dependency between $\lambda$ and $\beta$.

The mathematical model developed approximates the data obtained experimentally in a good manner; the differences obtained between the data from the theoretical model and the experimental ones for separating the seeds ($S_s$), which represent the most important quality index when harvesting cereals, are very small (1.29%).

This is due to the fact that at the beginning of the development of the mathematical model, several assumptions were made to eliminate the factors (which are not few) that each have a small influence on the threshing process; therefore, through the theoretically developed mathematical model, the process of separating the seeds is achieved almost completely (99.99%), and the experimentally verified is a little lower (98.71%) (as shown in Figure 5).

Figure 6 presents the variation of the distribution function of the seeds $S_d$ with the length of the threshing apparatus (comparative between the experimental and theoretical data), these having very close values, and Figures 7 and 8 show the variation of the distribution function of the free seeds ($S_l$), respectively, including the function of the distribution of unprocessed seeds ($S_n$), for the real data obtained from the experiments.

If it is followed on each section of the length of the threshing machine, respectively: 0.2/0.4/0.6,/0.8/1.0/1.2/1.4/1.6/1.8/2 m, $S_s + S_l + S_n = 100$%, practically; as the ears advance in the threshing apparatus, the percentage of separated seeds increases, and that of free seeds in straw ($S_l$) and unthreshed seeds ($S_n$) will decrease, tending towards zero (the difference to zero being represented by losses)—Figure 11.

Figures 9 and 10 show the values for the function of distribution of the separated seeds ($S_s$), respectively, for the function of the distribution density of the separated seeds ($S_d$), obtained as a result of the experiments for the 17 tests, based on the average values obtained for the function of the distribution of separated seeds (Figure 5; $S_{smed} = 98.71$%), respectively, for the function of the distribution density of the separated seeds (Figure 6, $S_{dmed} = 34.38$%).

**Author Contributions:** N.-V.V., S.-Ş.B., P.C., I.G., D.C., N.U., L.-D.P., L.P., G.M. and G.-C.T. have equal rights and have contributed evenly to the study design, collecting the data, measurements, modeling, data processing and interpretation of results and preparing the paper. All authors have read and agreed to the published version of the manuscript.

**Funding:** This research was supported by a grant of the Romanian Research and Innovation Ministry, through Programme 1—Development of the national research-development system, sub-programme 1.2—Institutional performance—Projects for financing excellence in RDI, contract no. 1 PFE/2021.

**Institutional Review Board Statement:** Not applicable.

**Informed Consent Statement:** Not applicable.

**Data Availability Statement:** Any data not reported in the paper will be provided on request.

**Acknowledgments:** All authors have equal rights and have contributed evenly to this paper. This work was supported by a grant of the Romanian Research and Innovation Ministry, through Programme 1—Development of the national research-development system, sub-programme 1.2—Institutional performance—Projects for financing excellence in RDI, contract no. 1 PFE/2021.

**Conflicts of Interest:** The authors declare no conflict of interest.

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
