# Peer review of "Contributions to the Mathematical Modeling of the Threshing and Separation Process in An Axial Flow Combine"

_agriculture, doi:10.3390/agriculture12101520_

Round 1

Reviewer 1 Report

The threshing process is a complex process. There are many factors affecting the threshing process, including not only the mechanical structure of the threshing device itself but also the properties of the harvesting object. The authors have made experiments and mathematical analyses on the threshing device by selecting several important factors. The content of this paper is rich and the derivation process is detailed. However, the research in this paper is aimed at the structure of a single threshing device. If the structure of the threshing device or the threshing gear bar is replaced, such as the closed threshing cylinder, the flexible gear bar, the nail gear bar, etc., is the model established in this paper still applicable? I hope the author will give an explanation. In addition, is this modeling method applicable to different crop varieties?

Some problems in the manuscript also need to be modified, as shown below, for the authors' reference.

1. In Section 3.6, the authors should analyze these figures in detail and point out the deviation between the theoretical modeling results and the actual values. In the Conclusions, the author should also give some quantitative descriptions.

2. The authors should carry out more tests under different working conditions, such as changing the rotation speed of the drum, to further verify the error between the model and the actual situation.

Reviewer 2 Report

The paper have presented a mathematical model that characterizes the process of threshing and separation from the threshing machine with axial flow of a thresher. The idea of the paper is novel, but there are some problems need to be revised and improved.

1.  In Section 2. Materials and Methods, the research object is not clearly described. The structure of each component should be shown in the figure. Test method should be described in detail and repeatable. It is suggested that the attached table explain the setting of test parameters.

2.  Many symbols in many formulas are not defined behind the formulas, which makes them really confusing, i.e. In function (6), what does ? represent? In function (6), what does ??(?), ??(?) represent? As there are too many variables in this manuscript, a nomenclature is recommended.

3.  The data in the paper should be carefully verified to avoid low-level errors , i.e. In line 67, a rotor with a diameter of 0.48 mm?

Round 2

Reviewer 1 Report

By reading the authors' response and the revised manuscript, I believe that the authors have revised the manuscript as required in terms of academic and technical aspects, and I recommend that the manuscript be accepted for publication in your journal.

Reviewer 2 Report

The author seriously answered the previous questions, and I think the revised manuscript has meet the publication requirements.